# Osmoregulation in the estuarine diamond-backed terrapin across a broad range of naturally occurring salinities

Jasmine Pierre, Brett M. Wilson and Amanda S. Williard*

## ABSTRACT

Diamond-backed terrapins are exposed to a broad range of salinities within their estuarine habitats, ranging from brackish water to full-strength seawater. The diamond-backed terrapin's ability to live exclusively in highly dynamic estuarine habitats has motivated experiments to explore the species' osmotic strategy; however, most of these experiments have taken place under laboratory conditions. The purpose of this project was to investigate the osmotic status of free-ranging diamond-backed terrapins during the active season in coastal southeastern North Carolina, USA. We collected blood samples from diamond-backed terrapins captured in salinities ranging from 9–39 psu and used linear models to assess the responses of blood osmolality, organic osmolytes, inorganic ions, and blood proteins to environmental and morphological variables. Results indicate that organic osmolytes play an important role in maintenance of body fluid homeostasis during exposure to high salinity. We found that salinity and body size have significant effects on blood variables which may reflect plasticity in osmotic strategy depending on demographic characteristics. We consider our results in the context of the energetic costs of maintaining osmotic homeostasis and the implications for diamond-backed terrapin resilience when exposed to altered salinity profiles due to changes in coastal land use and rising sea levels.

KEY WORDS: Organic osmolytes, Salinity, Sea level rise, Turtle, Urea

## INTRODUCTION

The diamond-backed terrapin (*Malaclemys terrapin* Schoepff, 1793) is an emydid turtle that lives in tidally influenced habitats along the East and Gulf coasts of the USA (Siegel and Gibbons, 1995). This species is the only North American turtle that is restricted in distribution to estuarine habitats (Dunson and Mazzotti, 1989). Monitoring programs have documented declines in diamond-backed terrapin populations due to a variety of coastal threats such as habitat loss, road mortality, and fisheries interactions (Siegel and Gibbons, 1995; Roosenburg et al., 1997; Dorcas et al., 2007; Crawford et al., 2013). Consequently, the diamond-backed terrapin is designated as Vulnerable by the International Union on the Conservation of Nature (IUCN) Red List (Roosenburg et al.,

Department of Biology and Marine Biology, University of North Carolina Wilmington, 601 S. College Rd., Wilmington, NC 28403, USA.

*Author for correspondence (williarda@uncw.edu)

(iD) A.S.W., 0000-0002-3286-2524

2019) and listed as a species of concern, threatened, or endangered by State management agencies throughout its range. Mean sea level rise (MSL) within the geographic areas occupied by diamond-backed terrapins is accelerating at a rate faster than that of global MSL (Kopp et al., 2015; Dangendorf et al., 2023), and diamond-backed terrapins are likely to experience significant alterations to their coastal habitats in the coming years. Successful long-term management of this unique estuarine reptile relies on an understanding of how the diamond-backed terrapin may respond to the environmental challenges presented by rising sea levels and associated impacts on coastal salinity profiles and habitat availability.

Temporal fluctuations in salinity of coastal marine and estuarine systems are caused by tidal cycles, seasonal changes in rainfall, and freshwater input from rivers; consequently, diamond-backed terrapins must regulate body fluid volume and composition in response to both acute and chronic changes in salinity (Dunson and Mazzotti, 1989). Diamond-backed terrapins maintain body fluid osmotic pressure at approximately one-third the osmotic pressure of seawater (Robinson and Dunson, 1976; Harden et al., 2015), which is generally lower than the osmotic pressure of the aquatic habitats in which they live. Due to the imbalance in osmotic pressure between diamond-backed terrapin body fluids and the external environment, both water and salts may passively move across permeable membranes according to osmotic or concentration gradients (Lillywhite and Evans, 2021). Maintenance of appropriate body fluid osmolality and volume requires regulation of water and salt uptake, the capacity to effectively excrete excess salt, and mechanisms to conserve water under highly variable salinity conditions. Diamond-backed terrapins, like other species of estuarine reptiles (Taplin, 1984; Lillywhite et al., 2008), take advantage of periodic access to low salinity water or freshwater in the form of rainfall to maximize water intake via drinking and enhance body water stores (Cowan, 1974; Davenport and Macedo, 1990), and reduce salt intake through hypophagy when exposed to high salinity with no access to freshwater (Davenport and Ward, 1993). The diamond-backed terrapin's lachrymal salt gland is activated when blood $Na^+$ levels exceeded 200 mM and facilitates excretion of ingested salt (Dunson and Dunson, 1975; Robinson and Dunson, 1976). With regards to water conservation, Robinson and Dunson (1976) illustrated that the diamond-backed terrapin has low integumentary permeability to water, thus minimizing passive exchange with the environment; however, diamond-backed terrapins may lose water across epithelial surfaces during prolonged exposure to high salinity, which could ultimately affect osmotic balance if other regulatory strategies are not employed (Bentley et al., 1967; Robinson and Dunson, 1976). For aquatic animals, an increase in body fluid osmotic pressure due to accumulation of organic osmolytes (e.g. urea, glucose) decreases the osmotic gradient between the animal's body fluids and the environment and promotes water retention (Withers, 1998; Yancey, 2005; Konno et al., 2006). Laboratory studies with diamond-backed terrapins illustrate that while variation in blood osmotic pressure at

lower salinities (up to half-strength seawater) is attributed to alterations in inorganic ion concentration ($Na^+$ and $Cl^-$), variation in osmotic pressure at higher salinities (half- to full-strength seawater) is primarily due to changes in blood urea levels (Gilles-Baillien, 1970). These results suggest that diamond-backed terrapins rely on accumulation of organic osmolytes to prevent water loss to the environment and maintain body fluid volume during exposure to high salinity.

The nature of compensatory osmotic responses to variation in environmental conditions depends on several factors, including the availability of behavioral options for osmoregulation (Harden et al., 2015; Williard et al., 2019). Disparities between significant osmotic responses exhibited by captive diamond-backed terrapins exposed to tightly controlled laboratory conditions and the relatively stable osmotic status of diamond-backed terrapins held under semi-natural conditions highlights this issue. For example, experiments with diamond-backed terrapins held in laboratory seawater tanks showed a significant increase in plasma urea over the course of overwintering (Gilles-Baillien, 1973) whereas diamond-backed terrapins overwintering in outdoor enclosures with suitable marsh habitat and tidal exposure to high salinity water (range 25–35 psu) exhibited stable plasma urea concentrations that were considerably lower than that of diamond-backed terrapins held in seawater tanks (Harden et al., 2015). Results for diamond-backed terrapins in semi-natural conditions suggest that behavioral adjustments are an important component of this species' osmotic strategy. Periodic access to freshwater via rainfall and the ability to utilize terrestrial mud habitats for winter burial provides diamond-backed terrapins with energetically efficient means of controlling body fluid composition during times when physiological and metabolic capacity are reduced due to temperature effects (Harden et al., 2015; Williard et al., 2019). Additional studies are warranted to explore the osmotic strategy of diamond-backed terrapins under natural conditions in which both behavioral and physiological adjustments may contribute to osmoregulation.

The purpose of this study was to investigate the osmotic responses of free-ranging diamond-backed terrapins to variations in environmental conditions during the active season when they are foraging and exploiting both terrestrial and aquatic habitats. We documented blood osmotic variables of diamond-backed terrapins captured across a broad range of environmental salinities and temperatures between April and October in southeastern NC, USA (Fig. 1). Behavioral plasticity of diamond-backed terrapins under natural conditions may provide them with additional options to maintain osmotic homeostasis compared with animals kept under controlled laboratory conditions. Nevertheless, if laboratory results are applicable to free-ranging terrapins, then we predict that higher environmental salinities will result in higher blood osmolality due primarily to an increase in urea. These adjustments would facilitate

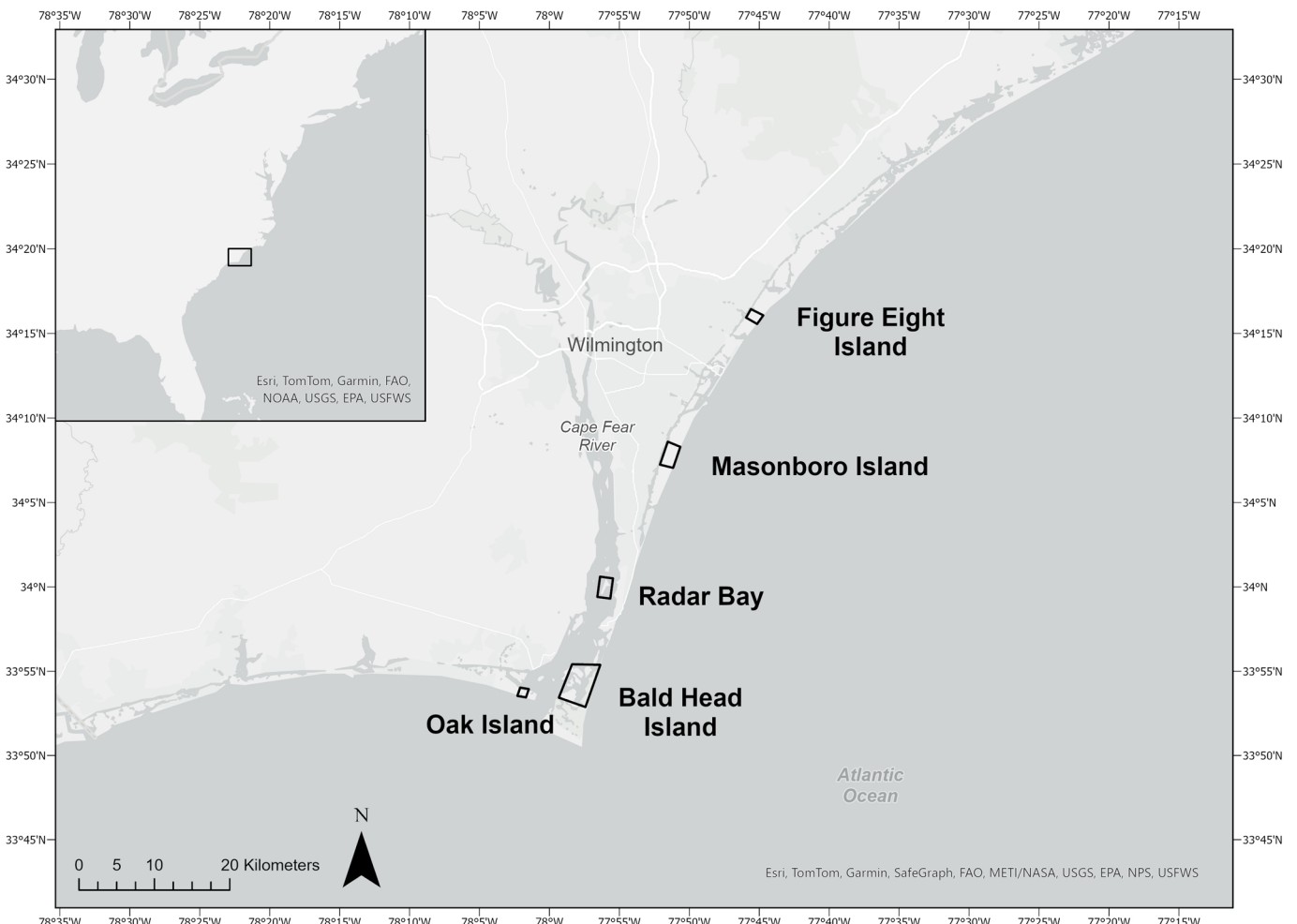

Fig. 1. Map indicating locations along the southeastern coastline of North Carolina, USA where diamond-backed terrapins were captured. From north to south: Figure Eight Island, Masonboro Island, Radar Bay, Bald Head Island, Oak Island.

water retention during periodic exposure to high salinities in dynamic estuarine systems. An understanding of the diamond-backed terrapin's osmoregulatory strategy will enhance our ability to assess how diamond-backed terrapins may respond to projected changes in coastal habitats.

## RESULTS

Blood samples were obtained from 51 diamond-backed terrapins, 40 captured in 2021 and 11 captured in 2022. Fourteen individuals with either low ($N$=5) or high ($N$=9) blood glucose values were excluded from data analysis to reduce the likelihood that individuals experiencing ill-health or excessive stress influenced the interpretation of results (Colon and Girolama, 2020). The 37 diamond-backed terrapins included in the statistical analyses were a mixture of immature and mature males ($N$=18) and females ($N$=19) that ranged in size from 97–175 mm straight carapace length (SCL) (Lovich et al., 2018). The salinity ranged between 9–39 psu and $T_W$ ranged between 18–30°C at diamond-backed terrapin capture sites. Environmental [salinity, water temperature ($T_W$), season], morphological [sex, SCL], and blood chemistry [osmolality, sodium ($Na^+$), chloride ($Cl^-$), potassium ($K^+$), urea, glucose, total protein] data for each diamond-backed terrapin used for statistical analyses are presented in Table S1. Descriptive statistics for each blood chemistry variable are reported in Table 1 and the results of a non-metric multidimensional scaling (NMDS) analysis to explore trends in blood chemistry profiles and identify significant explanatory variables are presented in Fig. 2. The stress value for the NMDS ordination for terrapin blood chemistry profiles was 0.131; stress values less than 0.2 indicate an acceptable fit (Dexter et al., 2018). A Shepard plot to assess the goodness of fit of the NMDS plot yielded an $R^2$ value of 0.922. Results of PERMANOVA tests showed weak but significant correlations between salinity ($P$=0.001, $R^2$=0.159), SCL ($P$=0.026, $R^2$=0.084), and season ($P$=0.008, $R^2$=0.099) and blood chemistry profiles. Neither $T_W$ ($P$=0.222, $R^2$=0.042) nor terrapin sex ($P$=0.104, $R^2$=0.057) had a significant effect on blood chemistry profiles.

Results from linear models to assess the effects of salinity, SCL and season on blood chemistry variables are provided in Table 2 and Fig. 3. Blood osmolality showed a significant positive relationship with salinity [estimate (95% confidence intervals) =0.047 (0.014, 0.079)], as did the organic osmolyte urea [0.194 (0.012, 0.376)]. A significant negative relationship with SCL was documented for the organic osmolytes urea [−0.237 (−0.386, −0.087)] and glucose [−0.120 (−0.225, −0.016)]. Blood $Na^+$ exhibited a positive

relationship with SCL [0.022 (0.001, 0.043)]. Explanatory variables did not have a statistically significant effect on blood $Cl^-$, $K^+$, or total protein.

## DISCUSSION

Diamond-backed terrapins rely on a suite of behavioral and physiological adjustments to maintain body fluid homeostasis over the broad range of salinities and temperature they encounter in coastal habitats (Harden and Williard, 2018). Previous laboratory studies provided evidence that the organic osmolyte urea plays a role in the diamond-backed terrapin's ability to exploit brackish water and full-strength seawater habitats (Gilles-Baillien, 1970), and our study with free-ranging diamond-backed terrapins supports that conclusion. Furthermore, results of our study highlight trends in blood chemistry variables that illustrate how differences in body size may affect osmotic strategy, particularly with regards to the degree of reliance on organic osmolytes for water conservation.

### Blood osmolality and organic osmolytes

Within the salinity range of 9–39 psu, we found significant positive relationships between environmental salinity and diamond-backed terrapin blood osmolality and urea. Organic molecules such as urea and glucose are used as osmoeffectors to maintain water balance between internal fluid compartments and to maintain appropriate body fluid volume in a wide variety of vertebrates (Jorgenson, 1997; Withers, 1998; Evans et al., 2004; Storey and Storey, 2017), including semi-aquatic amphibians (e.g. *Bufo viridis*, *Rana cancrivora*) and turtles (e.g. *Trionyx spiniferus*, *Pelodiscus sinensis*) that exploit brackish water habitats (Balinsky, 1981; Gordon et al., 1961; Katz, 1986; Lee et al., 2006; Seidel, 1975). The mechanism by which diamond-backed terrapins accumulate urea in the blood during exposure to high salinity has not been investigated but could involve urea retention or synthesis. Gilles-Baillien (1970) proposed that retention of urine in the bladder and reabsorption of water and urea across the vesicular epithelium of the bladder during periods of reduced freshwater availability and/or exposure to high salinity may provide diamond-backed terrapins with a means of regulating blood osmotic pressure and volume under osmotically challenging conditions. This is based on documented increases in urea in both blood and urine during prolonged exposure to high salinity, and the observation that diamond-backed terrapins held in freshwater void large volumes of urine that is hypoosmotic to blood, whereas diamond-backed terrapins held in seawater void isosmotic urine at one fifth the volume voided by freshwater-acclimated

**Table 1. Descriptive statistics of blood chemistry variables for 37 terrapins captured during the active season (April – October) in southeastern North Carolina, USA**

| Blood variable (units) | Mean±s.d. | Median (range) | Active season range[‡] | Overwintering range[§] |
|---|---|---|---|---|
| Osmolality (mOsm) | 339.6±30.6 | 335.5 (264.0–413.0) | - | 318.1–345.7 |
| $Na^+$ (mmol l$^{-1}$) | 157.2±10.0 | 156.0 (139.0–178.0) | 140.5–148.6 | 142.9–155.7 |
| $Cl^-$ (mmol l$^{-1}$) | 118.4±10.8 | 117.0 (101.0–140.0) | 105.4–116.5 | 103.7–110.1 |
| $K^+$ (mmol l$^{-1}$) | 4.3±0.5 | 4.2 (3.2–5.5) | 3.7–3.9 | 2.6–3.5 |
| Urea* (mmol l$^{-1}$) | 33.0±17.3 | 32.5 (8.9–90.0) | 13.2–31.7 | 30.6–44.9 |
| Glucose (mg dl$^{-1}$) | 85.6±29.2 | 79.0 (50.0–149.0) | 67.5–75.6 | 37.8–59.5 |
| Total Protein (g dl$^{-1}$) | 2.5±0.8 | 2.5 (1.0–5.2) | 2.3–3.0 | - |

*Urea values converted from reported BUN values using the equation urea=BUN *0.357.

[‡]Values from Ramirez (2014). Range of mean concentrations of blood variables measured in male and female diamond-backed terrapins (mean mass=352.0–1504.6 g) collected from three estuaries along the East coast of Texas, USA between April and November in 2015 and 2016. [§]Values from Harden et al. (2015). Range of monthly mean concentrations of blood variables measured in diamond-backed terrapins in estuaries along the East coast of NC, USA between November – April in 2010, 2011, and 2012. Data from adult female diamond-backed terrapins (mass=300–700 g) held in outdoor enclosures that included sub-tidal, low- and high-marsh habitat (semi-natural conditions) and free-ranging diamond-backed terrapins. Glucose values reported as mmol l$^{-1}$ converted to mg dl$^{-1}$ using the Eqn. 1 mmol l$^{-1}$=18.0182 mg dl$^{-1}$.

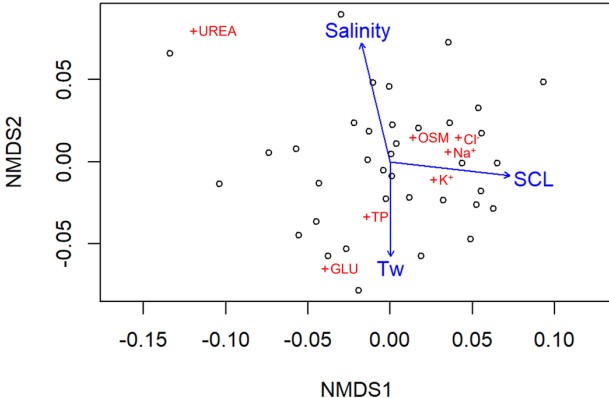

**Fig. 2. NMDS ordination map displaying the distribution and grouping of individual blood chemistry variables (+) relative to the ordination scores for blood chemistry profiles of 37 diamond-backed terrapins (○).** Continuous explanatory variables are overlaid onto the ordination plot as vectors (blue lines). OSM, Osmolality; GLU, glucose; TP, total protein.

diamond-backed terrapins (Bentley et al., 1967; Gilles-Baillien, 1970). Use of urine stored in the bladder as a potential source of water and balancing osmolytes has been proposed for other chelonians, such as desert tortoises (*Gopherus agassizii*), that regularly experience desiccating conditions and water deprivation in their natural habitats (Dantzler and Schmidt-Nielsen, 1966; Peterson, 1996). The significant increase in urea with an increase in salinity observed in diamond-backed terrapins may also be due to higher rates of urea synthesis, as has been proposed for salt-tolerant species in the family Trionychidae. Lee et al. (2006) found that Chinese softshell turtles exposed to 15 ppt salinity for 6 days increased urea synthesis 1.4 fold compared with urea synthesis rates in freshwater. Plasticity in rates of urea synthesis in response to variable environmental salinity has not yet been described in diamond-backed terrapins but is a topic worthy of further investigation as it may provide additional insight into the energetic costs of osmoregulation in this species.

### Body size and organic osmolytes
An interesting trend noted in our study was the inverse relationship between body size, as measured by SCL, and both urea and glucose. Previous research indicates that urea and glucose serve as cellular cryoprotectants for freeze-tolerant hatchling turtles, including diamond-backed terrapins (Costanzo et al., 2006). Diamond-back terrapin hatchlings that overwinter in terrestrial nests experience osmotic challenges when ice forms in their body fluids. Accumulation of urea and glucose prevent cellular dehydration, limit ice formation, and promote overwintering survival in this early life stage (Costanzo et al., 2006; Storey and Storey, 2017). These

same organic osmolytes may also serve as osmoeffectors and slow down rates of water loss to the environment in small free-living juvenile and adult diamond-backed terrapins under osmotically challenging conditions. Small diamond-backed terrapins may be less effective at passively regulating body fluid volume and composition due to their high surface area to volume ratio and high potential for convective exchange with the surrounding environment (Foley and Spotila, 1978). Dunson (1985) found an inverse relationship between body size and water efflux in diamond-backed terrapins acclimated to full-strength seawater, and a generally lower salinity tolerance in hatchlings compared with adult diamond-backed terrapins. Murphy et al. (2016) found a negative correlation between body size and evaporative water loss in four species of semi-aquatic turtles (*Deirochelys reticularia*, *Kinosternon subrubrum*, *Sternotherus oderatus*, and *Trachemys scripta scripta*) and suggest that body mass could be a useful predictor for assessing desiccation risk in turtles. If smaller turtles are at greater risk for desiccation this may be reflected in both behavioral and physiological aspects of their osmotic strategy. A stronger reliance on organic osmolytes to combat passive water exchange with the environment may reflect the higher desiccation risk for small diamond-backed terrapins.

### Regulation of inorganic ions
Osmotic and diffusion gradients for diamond-backed terrapins immersed in high salinity water would favor water efflux and an influx of ions across the epithelial surfaces (Robinson and Dunson, 1976; Harden and Williard, 2018; Lillywhite and Evans, 2021), which could contribute to an increase in body fluid osmolality. Ramirez (2014) noted a positive correlation between plasma $Na^+$ and environmental salinity at the time of capture for free-ranging diamond-backed terrapins in TX, USA; however, our results showed no significant positive relationship between either $Na^+$ or $Cl^-$ and salinity. Laboratory studies have documented an increase in $Na^+$ and $Cl^-$ with an increase in salinity for captive diamond-backed terrapins, but only at lower salinities. For example, Gilles-Baillien (1970) found a concurrent increase in plasma $Na^+$, $Cl^-$, and osmolality for diamond-backed terrapins transferred from freshwater to 50% seawater; but not for diamond-backed terrapins transferred from 50% seawater to 100% seawater. Diamond-backed terrapins may allow fluctuations in plasma $Na^+$ and $Cl^-$ levels below a threshold concentration but invoke regulatory mechanisms to stabilize $Na^+$ and $Cl^-$ concentrations during exposure to higher salinities. We documented a significant positive relationship between SCL and plasma $Na^+$ levels, which suggests that larger animals may be more tolerant of higher salt loads in their body fluids; however, hypernatremia has not been well-studied in diamond-backed terrapins and this topic requires additional investigation before conclusions can be drawn.

Regulation of inorganic ions may be accomplished by employing behaviors that reduce the potential for water and ion exchange

**Table 2. Results of the linear models to assess the effects of salinity, straight carapace length (SCL), and season on blood chemistry variables**

| Blood Variable | Intercept | Salinity (psu) | SCL (mm) | Season |
|---|---|---|---|---|
| Osmolality | **5.835 (5.796, 5.874)** | **0.047 (0.014, 0.079)** | −0.005 (−0.032, 0.021) | −0.018 (−0.084, 0.047) |
| Na⁺ | **5.081 (5.050, 5.111)** | 0.002 (−0.027, 0.024) | **0.022 (0.001, 0.043)** | −0.041 (−0.092, 0.010) |
| Cl⁻ | **4.803 (4.760, 4.847)** | 0.011 (−0.025, 0.047) | 0.028 (−0.002, 0.057) | −0.053 (−0.126, 0.019) |
| K⁺ | **1.659 (1.604, 1.714)** | 0.005 (−0.040, 0.050) | −0.003 (−0.040, 0.034) | −0.003 (−0.095, 0.089) |
| Urea | **3.327 (3.106, 3.548)** | **0.194 (0.012, 0.376)** | **−0.237 (−0.386, −0.087)** | −0.173 (−0.195, 0.542) |
| Glucose | **4.327 (4.172, 4.483)** | −0.028 (−0.155, 0.100) | **−0.120 (−0.225, −0.016)** | 0.181 (−0.077, 0.440) |
| Total Protein | **1.204 (1.085, 1.323)** | 0.027 (−0.071, 0.125) | 0.024 (−0.056, 0.105) | 0.041 (−0.157, 0.240) |

Estimates and 95% confidence intervals (CI) are provided for each blood chemistry variable. Response variables were log-transformed and continuous explanatory variables were standardized prior to incorporation in the models. Statistical significance is indicated by bold text.

Biology Open

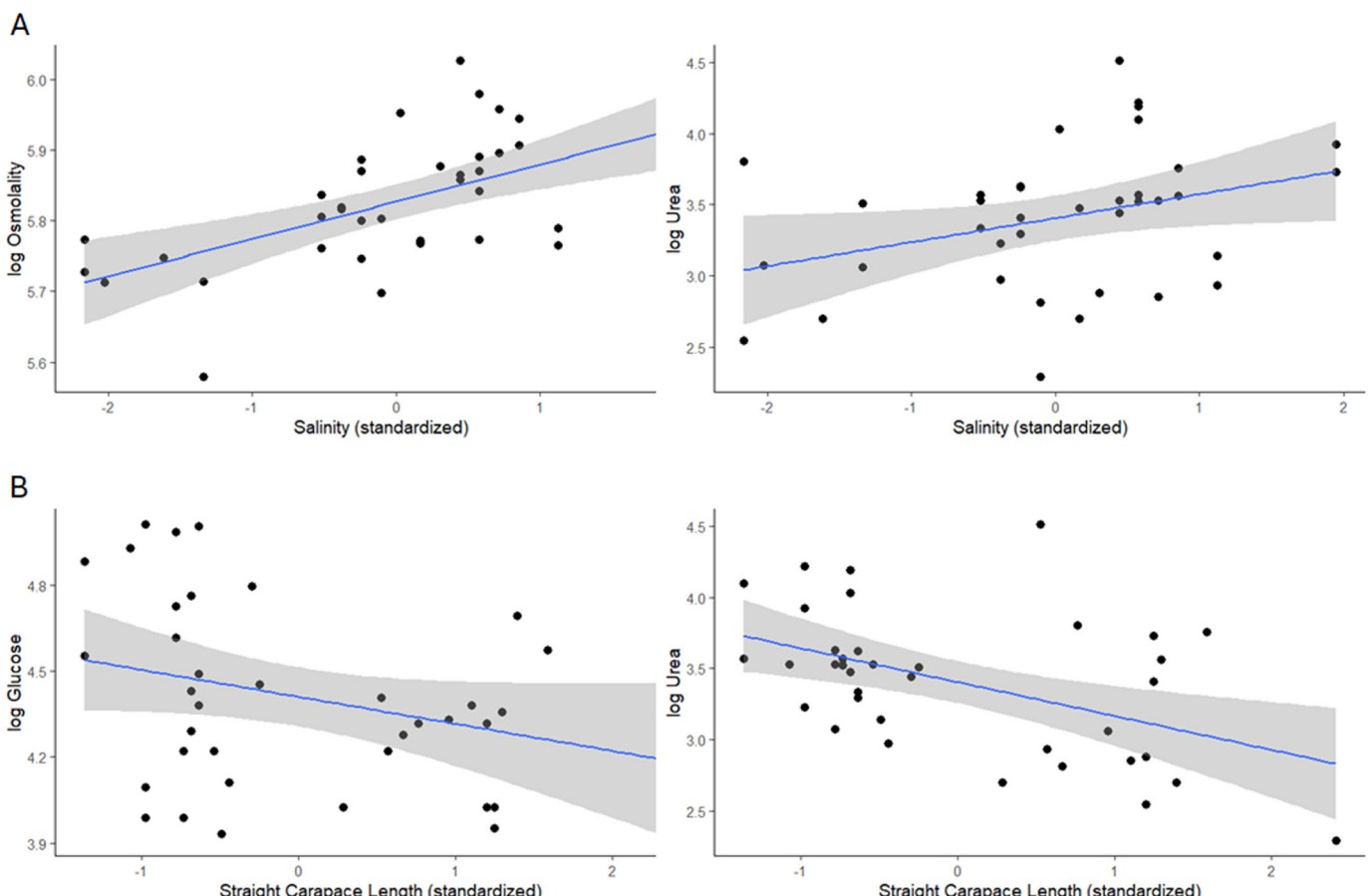

**Fig. 3. Graphs illustrating model output for log-transformed blood chemistry variables in relation to statistically significant explanatory variables.** Linear regression analysis indicated that there was a significant positive effect of salinity on osmolality [estimate (95% confidence intervals)=0.047 (0.014, 0.079)] and urea [0.194 (0.012, 0.376)] (A) and a significant negative effect of straight carapace length (SCL) on glucose [−0.120 (−0.225, −0.016)] and urea [−0.237 (−0.386, −0.087)] (B). Raw data for blood chemistry variables for diamond-backed terrapins (n=37) and explanatory variables are provided in Table S1.

with the environment, such as terrestrial mud burial or basking (Davenport and Magill, 1996), and/or energy-requiring physiological mechanisms to rid the body of excess salts, such as ion secretion via the extrarenal salt glands (Cowan, 1981). Experiments in which diamond-backed terrapins were salt-loaded with an injected NaCl solution showed that the salt gland is maximally activated when plasma $Na^+$ concentration exceeds 200 mmol $l^{-1}$ and salt gland activation was associated with an increase in $Na^+$ $K^+$ ATPase activity (Dunson and Dunson, 1975). That said, the salt gland may be activated at lower plasma $Na^+$ concentrations depending on the salinity acclimation regime; diamond-backed terrapins acclimated to freshwater under laboratory conditions exhibited an increase in salt gland secretions when injected with a salt solutions that raised plasma $Na^+$ to 160 mmol $l^{-1}$ (Cowan, 1981). We documented plasma $Na^+$ concentrations of 139–179 mmol $l^{-1}$ for free-ranging terrapins captured in salinities between 9–39; these values are similar to those previously reported for captive (Williard et al., 2019) and free-ranging diamond-backed terrapins (Table 1) and include values at which salt gland activation occurs (Dunson and Dunson, 1975; Cowan, 1981). Previous research suggests that active ion transport via the salt gland may play a larger role in the diamond-backed terrapin's osmotic strategy during exposure to a salinity challenge or transient increases in plasma $Na^+$ such as may occur while feeding in high-salinity environments (Gilles-Baillien, 1970).

Despite the evidence that salt gland activation is associated with an increase in energy use via $Na^+$ $K^+$ ATPase activity (Dunson and Dunson, 1975), the limited number of studies on diamond-backed terrapin energetics have not detected a significant effect of salinity on overall metabolic rate. Williard et al. (2019) detected no significant difference in oxygen consumption of fasted diamond-backed terrapins acclimated to salinities of 12 or 35. Likewise, Holliday et al. (2009) did not detect differences in metabolic rate of fasted diamond-backed terrapins acclimated to salinities ranging from 0–30; however, diamond-backed terrapins acclimated to high salinities exhibited slower growth rates compared with those acclimated to low salinities. Energetic trade-offs to accommodate increased metabolic costs of osmoregulation during prolonged exposure to higher salinities may impact other aspects of diamond-backed terrapin life history and ecology (Dunson and Dunson, 1975; Holliday et al., 2009; Williard et al., 2019). Behavioral adjustments, such as mud burial, and morphological features that reduce exchange of water and salts with the environment offer a more energetically efficient means of osmoregulation for diamond-backed terrapins (Davenport and Macedo, 1990; Davenport and Magill, 1996; Harden et al., 2015). In addition to the use of terrestrial habitats to slow down passive exchange of water and salts with the environment (Davenport and Magill, 1996), changes in feeding behavior may be used to regulate salt intake during exposure to high salinity. Several studies have documented a reduction in

food intake for diamond-backed terrapin exposed to high salinities (Dunson, 1985; Davenport and Ward, 1993; Holliday et al., 2009; Ashley et al., 2021; Williard et al., 2019), and a reduction in feeding with an increase in environmental salinity has also been documented in freshwater turtle species that periodically use brackish water habitats (Agha et al., 2018).

## Osmoregulation and anthropogenic stressors

For our study, we used a method of capture that is commonly employed in field studies with diamond-backed terrapins. Modified crab pots that permit access to air reduce the likelihood of injury and stress for captured diamond-backed terrapin. We did not measure corticosterone, and indicator of induction of the systemic stress response (Romero, 2004), in our study but Ramirez (2014) reported corticosterone levels of $4.7 \pm 1.0$ ng ml$^{-1}$ (mean$\pm$s.d.) for diamond-backed terrapins removed from crab pots with a soak time of 24 h. Corticosterone values for diamond-backed terrapins removed from crab pots are slightly higher than values reported for diamond-backed terrapins captured by hand and sampled immediately ($\leq 4$ ng ml$^{-1}$, Winters et al., 2016) but comparable to or lower than values for diamond-backed terrapins minimally handled for 30–60 min ($\leq 8$ ng ml$^{-1}$, Winters et al., 2016) and sea turtles removed from entanglement nets (mean values 6.2–24.7 ng ml$^{-1}$, Gregory and Schmid, 2001) gillnets (20.8$\pm$16.5 ng ml$^{-1}$, Snoddy et al., 2009), or longline fishing gear (4.3$\pm$3.6 ng ml$^{-1}$, Williard et al., 2015). Elevated levels of corticosterone may cause an increase in blood glucose levels (Romero, 2004); based on published values for blood glucose in minimally handled wild-caught chelonians (Anderson et al., 2011; Espinoza-Romo et al., 2018, Harden et al., 2018; Stacy et al., 2018; Perrault et al., 2020), we excluded individuals with glucose levels $>150$ mg dl$^{-1}$ to reduce the likelihood that stress influenced our interpretation of blood chemistry results. Sea turtles and marine mammals exposed to acute stressors exhibit a positive association between corticosterone and aldosterone, a mineralocorticoid hormone that mediates Na$^+$ reabsorption and water retention and plays an important role in body fluid homeostasis in vertebrates (Champagne et al., 2015; Innis et al., 2024). The concurrent increase in corticosterone and aldosterone during long-term exposure to environmental stressors could have implications for salt and water balance in the estuarine diamond-backed terrapin, and this relationship warrants further investigation with regards to anthropogenic disturbances.

An understanding of the diamond-backed terrapin's osmotic strategy and adaptability to highly dynamic estuarine environments is crucial for predicting how this species of conservation concern may respond to the challenges presented by climate change and rising sea levels (Lamont et al., 2025). Sea level rise along the East and Gulf coasts of the USA is occurring at a faster rate than the global mean (Dangendorf et al., 2023), with important implications for terrestrial and aquatic organisms that use coastal habitats in this region (Woodland et al., 2017; Agha et al., 2018, 2019). For example, at our study location in the lower Cape Fear River in southeastern North Carolina tidal influence has extended upstream due to both sea level rise and dredging and this has resulted in upriver migration of saltmarsh and alterations to vegetation along tidal creeks (Magolan and Halls, 2020). Modelling exercises to assess habitat availability for diamond-backed terrapins in Chesapeake Bay, USA under different sea level rise scenarios predicted significant alterations in, and loss of, terrestrial and brackish water habitats that could affect multiple aspects of diamond-backed terrapin life history (Woodland et al., 2017). The semi-aquatic diamond-backed terrapin's ability to exploit highly variable estuarine environments depends on

a balance of energy-requiring physiological adjustments and energetically efficient behavioral modifications employed in both terrestrial and aquatic habitats to cope with salinity challenges and maintain internal homeostasis. Some behavioral components of the diamond-backed terrapin's osmotic strategy, such as reducing food intake under high salinity conditions, are highly effective in the short term but have energetic consequences (i.e. reduced energy acquisition) and are unsustainable over the long term. Likewise, physiological mechanisms to excrete excess salt or accumulate organic osmolytes may incur energetic costs and/or require a reallocation from other energy-requiring processes such as growth (Holliday et al., 2009), which may affect demographic characteristics of populations. The ability of the diamond-backed terrapin to adapt to changing coastal conditions in the future will depend on finding the appropriate balance of behavioral and physiological adjustments to meet osmotic challenges in the context of other life history requirements and habitat availability.

## MATERIALS AND METHODS
### Field sampling

Diamond-backed terrapins were collected at five sites in the lower Cape Fear River and along the Intracoastal Waterway in southeastern North Carolina (Fig. 1) from April to October 2021 and April to July 2022. The time of capture was categorized as occurring during the breeding (April–June) or post-breeding (July–October) season. Seasonal categorization was determined based on the patterns observed when monitoring terrapin nesting season in previous studies (Burger and Montevecchi, 1975; Butler et al., 2004; Seigel, 1980). A recent study analyzing genetic characteristics of diamond-backed terrapins in southeastern North Carolina showed no significant genetic structuring between the five sampling sites (B.M. Wilson, personal communication); consequently, capture location was not used as a factor in analyses of osmotic responses.

All field procedures used to capture and collect blood samples and biological data from diamond-backed terrapins were approved by the University of North Carolina Wilmington Institutional Animal Care and Use Committee (IACUC Protocol A2122-011). Diamond-backed terrapins were captured using standard commercial crab traps ($61 \times 61 \times 61$ cm) fitted with a wire chimney (122 cm height$\times$30.5 cm diameter) to allow entrapped diamond-backed terrapins access to air during all stages of the tidal cycle (Chavez and Williard, 2017). The modified crab pots were stabilized by iron bars along the base and PVC pipes driven into adjacent mud to prevent the pot from tipping over due to wave action or extreme weather events. Crab pots were checked every 24–48 h ($33.9 \pm 11.6$ h, $\bar{X} \pm$s.d.) to ensure that captured diamond-backed terrapins were removed in a timely manner. Salinity and water temperature ($T_W$) were measured each time the pot was checked.

Straight carapace length (SCL, mm) was measured with calipers and each captured diamond-backed terrapin was marked externally with unique notchings on the marginal scutes that corresponded to a three-letter code (Dorcas et al., 2007). A 1–2 ml blood sample was collected from the subcarapacial sinus of each diamond-backed terrapin using a 23G1″ needle and 3cc syringe. Blood samples were transferred immediately to lithium heparin coated Vacutainers to prevent coagulation and stored on ice for transfer to UNCW main campus. In the lab, blood samples were centrifuged at 3000 $g$ for 5 min to separate plasma from blood cells. Plasma was transferred to cryogenic storage vials and stored at $-80°$C.

### Blood analyses

A vapor pressure osmometer (Vapro 5520, Wescor Inc., Logan, UT, USA) was used to measure blood osmolality for a 10 µl sub-sample of plasma. Calibration standards of 290 and 1000 mOsm were run every ten samples to ensure accuracy of measurements. A 300 ml sub-sample of plasma was sent to a veterinary diagnostic lab (Antech Diagnostics Inc., Fountain Valley, CA, USA) for a Reptile Comprehensive Chemistries analysis. This panel

Biology Open

returns results for major electrolytes [sodium (Na$^+$, mmol l$^{-1}$), chloride (Cl$^-$, mmol l$^{-1}$), potassium (K$^+$, mmol l$^{-1}$)], organic osmolytes [blood urea nitrogen (BUN, mg dl$^{-1}$), glucose (mg dl$^{-1}$)], and blood proteins (total protein, g dl$^{-1}$). BUN values were converted to urea (mmol l$^{-1}$) for statistical analyses and comparison with published values for diamond-backed terrapins according to the following equation: [urea (mmol l$^{-1}$)= BUN*0.357]. Diamond-backed terrapins with blood glucose values <50 mg dl$^{-1}$ (hypoglycemia) or >150 mg dl$^{-1}$ (hyperglycemia) were excluded from statistical analysis to reduce the likelihood that individuals experiencing ill health or excessive stress influenced the interpretation of results (Colon and Girolama, 2020).

## Statistical analyses

The results from the blood osmolality measurements and veterinary diagnostic blood chemistry panel were statistically analyzed using R software (version 4.1.3). A non-metric multidimensional scaling (NMDS) analysis using Bray Curtis distance measure was performed to explore trends in blood chemistry profiles and identify explanatory variables that would be suitable for linear models. The *vegan* package in R was used to generate a distance matrix for the blood chemistry profiles of individual diamond-backed terrapins (Zuur et al., 2007). The output for the NMDS analysis generated an ordination map that displayed similarities in blood chemistry profiles in relation to the environmental (salinity, $T_W$, and season) and morphological (SCL and sex) explanatory variables. The NMDS analysis generated a stress value to quantitatively determine the goodness of fit of the ordination plot. Permutational Analysis of Variance (PERMANOVA, adonis2 function in *vegan* package in R) was used to assess whether a given explanatory variable had a significant effect on blood profiles. The continuous explanatory variables were displayed on the NMDS plot as vectors.

Individual blood chemistry variables (osmolality, Na$^+$, Cl$^-$, K$^+$, urea, glucose, total protein) were log-transformed and linear models were used to analyze the explanatory variables and factors that affect each blood chemistry variable. Based on the results of the NMDS and PERMANOVA analysis, continuous explanatory variables included in the linear models were salinity and SCL; season was included as a categorical factor. Continuous numeric explanatory variables were normalized using Z-score standardization. Significance was assessed based on exclusion of zero from the 95% confidence intervals around the model estimate. Code used for statistical analyses is provided in the Supplementary Information.

## Acknowledgements
Joe Facendola and Colin Smoot were instrumental in field work to collect diamond-backed terrapin blood samples and environmental data. Kristi Mitchell and Michael Tift provided laboratory logistics support. Lori Schweikert and Brian Arbogast provided comments on early drafts of the manuscript.

## Competing interests
The authors declare no competing or financial interests.

## Author contributions
Conceptualization: A.S.W.; Formal analysis: J.P., B.W., A.S.W.; Methodology: J.P., B.W., A.S.W.; Project administration: A.S.W.; Resources: A.S.W.; Supervision: A.S.W.; Visualization: J.P., B.W., A.S.W.; Writing – original draft: J.P.; Writing – review & editing: B.W., A.S.W.

## Funding
This work was supported by The North Carolina Division of Marine Fisheries (CW19138 to A.S.W.), a University of North Carolina Wilmington Cahill Grant (A.S.W.), and The Diamondback Terrapin Working Group Student Research Grant (J.P.). Open Access funding provided by University of North Carolina Wilmington (UNCW) Library. Deposited in PMC for immediate release.

## Data and resource availability
All relevant data and details of resources can be found within the article and its Supplementary Information.

## Peer review history
The peer review history is available online at https://journals.biologists.com/bio/lookup/doi/10.1242/bio.062072.reviewer-comments.pdf

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
