## [Peer Review File · Biology Open]

Osmoregulation in the estuarine diamond-backed terrapin across a broad range of naturally occurring salinities

Jasmine Pierre, Brett Wilson and Amanda Southwood Williard
DOI: 10.1242/bio.062072

Editor: Lewis Halsey

Review timeline

Original submission:	8 May 2025
Editorial decision:	10 May 2025
Resubmission:	16 May 2025
Editorial decision:	26 May 2025
First revision received:	29 May 2025
Accepted:	2 June 2025

Original submission

Decision letter

MS ID#: bio.62058

MS Title: Osmoregulation in the estuarine diamond-backed terrapin across a broad range of naturally occurring salinities

Authors: Amanda Williard; Jasmine Pierre; Brett Wilson

Dear Dr Williard,

I have now reached a decision on the above manuscript.

The reviewer reports are shown at the bottom of this email or can be accessed, together with a copy of this decision letter, by going to:

Review of your article has raised important concerns that are significant enough to prevent me from accepting it for publication. I am sorry to write with this disappointing news; however, I am sure that you appreciate that the conclusions of your research must be seen by the wider community to be fully supported by the data. Most importantly, the blood glucose levels are high enough to suggest that many of the turtles were stressed, generating a highly likely confound to the other physiological variables measured. The reviewer suggests that, possibly, enough turtles might not have very high glucose levels to represent a sufficient sample size for an unconfounded data set. (Indeed, moreover just possibly there could be an interesting comparison of the physiological responses between 'unstressed' and 'stressed' animals, and if so possibly as a separate manuscript).

Therefore, should you be able to carry out all the analysis suggested by the referees, and with a sufficient sample size, then I would be happy to see the manuscript again, as a new submission.

Reviewer 1

Comments for the author

The diamond-backed terrapin belongs to an exclusive and small group of reptiles that are tolerant to high salinity. It is of considerable interest therefore to understand the mechanisms that confer this tolerance, and this species has - not surprisingly - been studied over the past 50 years. Most of these studies have been performed as classic laboratory-based experiments, and the present manuscript is therefore important in providing values obtained on free-living turtles in their natural habitat.

The study reports on plasma composition of 51 specimens caught and sampled during two consecutive summers. in the summer of two years. Plasma was measured using reliable standard techniques.

The study seeks to establish correlations between water salinity and plasma composition, but as major drawback, there is no assessment on the duration of exposures. This is not easy to resolve with the current experimental design.

Overall, the manuscript is well-written and provides a good and detailed account of the earlier studies on this species. The manuscript is therefore interesting to read, and I only have one major concern as well as some minor comments for the authors to consider.

My major concern is the conspicuous lack of data presentation. Virtually all data - with the exception of Table 2 that report on ranges - are shown as log transformed and derived statistical analyses that provides very little physiological insight for the reader. I strongly encourage the existing statical presentations to be placed in the supplementary material, but that the authors create figures showing the relationships of osmolarity, the major ions, glucose and urea as a function of the salinity of the water. In this manner, a physiologist can evaluate the data.

other suggestions.

1. While well-written, the manuscript is rather long given the rather small data set. I believe this may deter readers from finishing reading the manuscript and I strongly suggest that the authors consider shortening the speculative section.
2. The samples were taken from the subcarapacial sinus, which is a standard approach. It is my personal experience that samples often are contaminated by lymph. Is this a concern?
3. The sampling also involves stress and the authors acknowledge that such stress is likely to have elevated plasma glucose. Stress may also explain the very high plasma potassium levels 7 mM reported in line 311). Hemolysis may also have contributed .
4. Much of the interpretation on the adaptive role of reducing water and ion fluxes centers on the assumption that osmoregulation is energetically expensive. Is there any evidence for this. In lines 304-9 you specifically cite papers showing no influence of water salinity on metabolism.
5. On line 63-65 it may be relevant to include the perspective that sea snakes, although normally considered quite well-adapted to the marine environment, als
6. Line 110 and elsewhere (eg line 233). It is a bit odd that water salinity is reported as psu. This is of course okay, but it would help the reader if values in mM were to be included as well.
7. Line 179. Units should be provided
8. Line 265, no need for decimals
9. Line 304. Is an abbreviation for water salinity (Tw) needed. It would be easier for the reader if spelled out

Reviewer 2:

Comments for the author

To better understand the osmoregulatory challenges of diamond-back terrapins that live in habitats with variable salinities, the authors have carried out the first sampling of blood from wild diamond-back terrapins from different habitats and correlated various blood parameter to environmental factors such as salinity and temperature.

I do have one major problem with the study:

We all know that studying stressed animals should be avoided as stress can affect virtually every physiological variable studied. Both the trapping of the terrapins (some animals may have been in the traps for up to 48 hours) and handling of the terrapins could induce stress that in turn can have strong effects on the blood variables measured. The authors acknowledge the value of cortisol measurements to ascertain that the animals are not overly stressed when captured and sampled. They spend half a page on discussing this, so it is puzzling and unfortunate that cortisol was not measured in the present study as it could have been used to exclude overly stressed animals from the analyses.

This leads me to my main methodological worry about the present study. The authors did measure glucose. High glucose, like high cortisol, is a hallmark of stress in animals. A major effect of stress hormones like cortisol and adrenaline is to elevate blood glucose (the fight-or-flight response). The mean glucose level reported in the study is clearly very high (101 mg/dl) in relation to previous studies on captive terrapins, where they range from 38 to 76 mg/dl (data given in Table 1). Indeed, the range reported in the present study appears exceptional, from 26 to 282 mg/dl. It is possible that there were relatively few animals that showed exceptionally high glucose levels, as indicated by the median glucose level (81 mg/dl) that is lower than the mean level. In my opinion the high and variable glucose levels preclude publishing the study. However, it could possibly be saved if individuals with very high glucose levels are excluded. Here, the previously published active-season range for this species (Ramirez (2014)) could be used to decide on a cutoff value that can be used to exclude animals that are likely to be highly stressed.

Other comments:

The introduction gives a good overview of the present knowledge of the osmoregulatory challenges and solutions for these terrapins. However, it is 4 pages long and more like a review of the subject and should be shortened substantially with focus of aspects directly relevant to the present study.

Like the Introduction, the Discussion is clearly too long and includes aspects that are not directly relevant to the data obtained in the present study. This includes discussions of water storage in the bladder (which presently was not studied), desert tortoises (which lives in completely different habitats). Also the lengthy discussion of urea handling is not needed.

Reviewer's Responses to Questions

Experimental quality

Does each figure have the proper controls?

If 'No', please indicate reasons in Comments for Author box below.

Reviewer #1:

- Yes

Reviewer #2:

- No

Were the data analyzed using appropriate statistical tests?

If 'No', please indicate reasons in Comments for Author box below.

Reviewer #1:

- Yes

Reviewer #2:

- Yes

Reproducibility

Were experiments performed using adequate number of biological replicates?

If 'No', please indicate reasons in Comments for Author box below.

Reviewer #1:

- Yes

Reviewer #2:

- Yes

Does the methods section provide sufficient detail to permit reproducibility?

If 'No', please indicate reasons in Comments for Author box below.

Reviewer #1:

- Yes

Reviewer #2:

- Yes

Completeness

Are the manuscript's conclusions supported by the data?

If 'No', please indicate reasons in Comments for Author box below.

Reviewer #1:

- Yes

Reviewer #2:

- Yes

Scholarship

Do the authors cite and discuss the merits of data that would argue for and against their conclusion?

If 'No', please indicate reasons in Comments for Author box below.

Reviewer #1:

- Yes

Reviewer #2:

- No

Does the manuscript title & abstract accurately reflect the contents of the manuscript, without hyperbole?

If 'No', please indicate reasons in Comments for Author box below.

Reviewer #1:

- Yes

Reviewer #2:

- Yes
-

Resubmission
MS ID# bio.062072

Author response to previous reviewers' comments

Reviewers' comments

Reviewer 1: The diamond-backed terrapin belongs to an exclusive and small group of reptiles that are tolerant to high salinity. It is of considerable interest therefore to understand the mechanisms that confer this tolerance, and this species has - not surprisingly - been studied over the past 50 years. Most of these studies have been performed as classic laboratory-based experiments, and the present manuscript is therefore important in providing values obtained on free-living turtles in their natural habitat. The study reports on plasma composition of 51 specimens caught and sampled during two consecutive summers. in the summer of two years. Plasma was measured using reliable standard techniques. The study seeks to establish correlations between water salinity and plasma composition, but as major drawback, there is no assessment on the duration of exposures. This is not easy to resolve with the current experimental design. Overall, the manuscript is well-written and provides a good and detailed account of the earlier studies on this species. The manuscript is therefore interesting to read, and I only have one major concern as well as some minor comments for the authors to consider. My major concern is the conspicuous lack of data presentation. Virtually all data - with the exception of Table 2 that report on ranges - are shown as log transformed and derived statistical analyses that provides very little physiological insight for the reader. I strongly encourage the existing statical presentations to be placed in the supplementary material, but that the authors create figures showing the relationships of osmolarity, the major ions, glucose and urea as a function of the salinity of the water. In this manner, a physiologist can evaluate the data. other suggestions.

We agree that it would be useful for the reader to have the raw data for individual terrapins and have provided a revised Table S1 in the Supplementary Data that includes environmental, morphological, and blood biochemistry data for each individual terrapin used in the statistical analysis for this study. The descriptive statistics for the raw data are provided in Table 1 within the main body of the manuscript, along with values from other studies with wild terrapins for comparison. We chose to present log-transformed response variables and standardized continuous explanatory variables in Figure 3 as these graphs are showing the output of the linear models that were used to assess significance of effects. The graphs show a direct representation of the model inputs. 1.

While well-written, the manuscript is rather long given the rather small data set. I believe this may deter readers from finishing reading the manuscript and I strongly suggest that the authors consider shortening the speculative section.

Thank you for this suggestion. We have streamlined the Introduction and Discussion. 2.

The samples were taken from the subcarapacial sinus, which is a standard approach. It is my personal experience that samples often are contaminated by lymph. Is this a concern?

We took detailed notes on the condition of blood samples upon collection. Samples for which we noted lymph draw prior to hitting the blood vessel were excluded from analysis.. The sampling also involves stress and the authors acknowledge that such stress is likely to have elevated plasma glucose. Stress may also explain the very high plasma potassium levels 7 mM reported in line 311). Hemolysis may also have contributed . The results provided by clinical laboratory indicated whether sample exhibited hemolysis; we did not include samples for which hemolysis could have impacted results. As requested by Editor and the second Reviewer, we applied a glucose cutoff criterion to exclude animals from our statistical analysis that may have been overly stressed. A detailed explanation of how we established glucose criterion is provided in the response to Reviewer 2 (please see below).

Much of the interpretation on the adaptive role of reducing water and ion fluxes centers on the assumption that osmoregulation is energetically expensive. Is there any evidence for this. In lines 304-9 you specifically cite papers showing no influence of water salinity on metabolism.

On line 63-65 it may be relevant to include the perspective that sea snakes, although normally considered quite well-adapted to the marine environment, als

This is a great point. I included a more general statement to include snakes and crocs: “Periodic access to low salinity water or freshwater in the form of rainfall enhances marine and estuarine reptiles’ ability to tolerate high salinity habitats (Taplin 1984; Lillywhite et al. 2008), and diamond-backed terrapins have been observed to selectively drink water with salinity < 20 psu (Cowan 1974; Davenport and Macedo 1990).”

Line 110 and elsewhere (eg line 233). It is a bit odd that water salinity is reported as psu. This is of course okay, but it would help the reader if values in mM were to be included as well.

I believe that it is most common to report mM for laboratory studies and environmental salinity as o/oo or psu for field studies

Line 179. Units should be provided We made this change as requested. 8. Line 265, no need for decimals

We made this change as requested.

Line 304. Is an abbreviation for water salinity (Tw) needed. It would be easier for the reader if spelled out

Tw was defined as water temperature earlier in the text.

Reviewer 2: To better understand the osmoregulatory challenges of diamond-back terrapins that live in habitats with variable salinities, the authors have carried out the first sampling of blood from wild diamond-back terrapins from different habitats and correlated various blood parameter to environmental factors such as salinity and temperature.

I do have one major problem with the study:

We all know that studying stressed animals should be avoided as stress can affect virtually every physiological variable studied. Both the trapping of the terrapins (some animals may have been in the traps for up to 48 hours) and handling of the terrapins could induce stress that in turn can have strong effects on the blood variables measured. The authors acknowledge the value of cortisol measurements to ascertain that the animals are not overly stressed when captured and sampled. They spend half a page on discussing this, so it is puzzling and unfortunate that cortisol was not measured in the present study as it could have been used to exclude overly stressed animals from the analyses. This leads me to my main methodological worry about the present study. The authors did measure glucose. High glucose, like high cortisol, is a hallmark of stress in animals. A major effect of stress hormones like cortisol and adrenaline is to elevate blood glucose (the fight-or-flight response). The mean glucose level reported in the study is clearly very high (101 mg/dl) in relation to previous studies on captive terrapins, where they range from 38 to 76 mg/dl (data given in Table

1). Indeed, the range reported in the present study appears exceptional, from 26 to 282 mg/dl. It is possible that there were relatively few animals that showed exceptionally high glucose levels, as indicated by the median glucose level (81 mg/dl) that is lower than the mean level. In my opinion the high and variable glucose levels preclude publishing the study. However, it could possibly be saved if individuals with very high glucose levels are excluded. Here, the previously published active-season range for this species (Ramirez (2014)) could be used to decide on a cutoff value that can be used to exclude animals that are likely to be highly stressed.

We appreciate the Reviewer's insight and agree that a more careful selection of individual terrapins to use in the analysis would be appropriate given the potential influence of capture stress on blood biochemistry. While the corticosterone levels of terrapins captured in modified crab pots were not exceptionally high in previous studies (Ramirez 2014 - MSc Thesis), the blood glucose levels we documented did show a wide range of values and extremely high values could indicate capture stress. We will use criteria based on published literature to remove individual terrapins that have excessively high or low glucose levels to ensure that overly stressed animals are not included in the statistical analysis, as suggested by the Reviewer.

Variation in blood glucose values in turtles can occur due to a variety of reasons, including nutritional status, reproductive and overwintering status, and stress. While we think that eliminating some individual terrapins from our analysis due to extremely high or low values is appropriate, we also think the analysis should accommodate the wide range of values that have been reported for minimally handled, healthy turtles in the veterinary and biological sciences literature. Although there is a paucity of data on blood biochemistry of wild-caught diamondbacked terrapins, there are numerous published studies that report values for blood biochemical variables in other turtles. Veterinary clinical studies with a wide range of tortoises and freshwater turtles indicate that severe hypoglycemia (defined as $< 30 \text{ mg}\cdot\text{dl}^{-1}$) and moderate ($> 150 \text{ mg}\cdot\text{dl}^{-1}$) to severe hyperglycemia ($> 180 \text{ mg}\cdot\text{dl}^{-1}$) are associated with poorer prognosis for chelonian patients (Colon and Girolamo 2020). Numerous studies documenting blood biochemistry of captive (rehabilitated) or minimally-handled wild-caught sea turtles have been published in recent years. The majority of blood glucose values documented for terrapins in our study fall well within the range of published values for sea turtles in captivity or captured by hand/dipnet or in trawls of ≤ 30 min (please see Stacy et al. 2018 for a comparison of values from multiple studies with North Atlantic loggerhead turtles). Based on the extensive literature, $150 \text{ mg}\cdot\text{dl}^{-1}$ has been proposed as a threshold for hyperglycemia in sea turtles (Perrault et al. 2020).

We have compiled a table with values and references to provide you with the typical values for glucose ($\text{mg}\cdot\text{dl}^{-1}$) in sea turtles and other chelonids that we are using to determine criteria for data inclusion in our terrapin analysis.

Species	Capture technique	Mean \pm SD	Range	Reference
Olive ridley (Lepidochelys olivacea)	Hand-caught on foraging grounds	122.6 \pm 34.8	62 - 224	Espinoza-Romo et al. 2018
Kemp's ridley (Lepidochelys kempii)	< 30 min trawl	†122.4 \pm 19.8	84.6 - 165.6	Perrault et al. 2020
Kemp's ridley (Lepidochelys kempii)	Captive (rehabilitated cold-stun)	†122.4 \pm 14.4	104.4 - 138.6	Innis et al. 2007
Loggerhead (Caretta caretta)	Hand-caught on foraging grounds	114.0	88.0 - 174.0	Stacy et al. 2018
Loggerhead (Caretta caretta)	< 30 min trawl	95.0	54.0 - 171.0	Stacy et al. 2018
Green (Chelonia mydas)	Captured in pound nets (low stress)	124	97 - 218	Anderson et al. 2011
Ornate box turtle (Terrapene ornate)	Hand-caught in natural habitat	71.8	22 - 154	Harden et al. 2018

Gopher tortoise (Gopherus polyphemus)	Captive (healthy)	74.5	55 - 128	Taylor and Jacobson 1982
Amazon freshwater turtle (Podocnemis expansa)	Captive (healthy)	91.3 ± 17.7	63.3 - 134.6	Oliveira-Junior et al. 2009
New Guinea snapping turtle (Elseya novaguineae)	Captive (healthy)	Male - 85 (median) Female - 122 (median)	Male - 52 - 177 Female - 23 - 215	Anderson et al. 1997

† Values for glucose in $\text{mmol}\cdot\text{L}^{-1}$ converted to $\text{mg}\cdot\text{dl}^{-1}$ using the formula: $\text{mg}\cdot\text{dl}^{-1} = \text{mmol}\cdot\text{L}^{-1} * 18$

Based on the published literature, we propose using only individual terrapins with blood glucose values between 50 - 150 $\text{mg}\cdot\text{dl}^{-1}$ ($n = 37$) to reduce the likelihood that stress or ill-health influences the data analysis and interpretation of results. We ran NMDS and linear models on the dataset that includes only the 37 terrapins that fit this criteria (glucose 50 - 150 $\text{mg}\cdot\text{dl}^{-1}$) and the results are mostly the same as those reported in the original paper, with the exception that water temperature was no longer a significant factor to be included in models (based on NMDS PERMANOVA) and there was a marginally significant effect of body size (SCL) on sodium. All other results were the same, including the primary findings of 1) a statistically significant positive relationship between salinity and blood osmolality and urea, and 2) a statistically significant inverse relationship between body size (SCL) and urea.

Other comments: The introduction gives a good overview of the present knowledge of the osmoregulatory challenges and solutions for these terrapins. However, it is 4 pages long and more like a review of the subject and should be shortened substantially with focus of aspects directly relevant to the present study.

Thank you for this suggestion. We have streamlined the Introduction such that it is now just 3 pages.

Like the Introduction, the Discussion is clearly too long and includes aspects that are not directly relevant to the data obtained in the present study. This includes discussions of water storage in the bladder (which presently was not studied), desert tortoises (which lives in completely different habitats). Also the lengthy discussion of urea handling is not needed. We have streamlined the Discussion. Water storage in bladder is only discussed in context of previous studies with terrapins.

Resubmission

First decision letter

MS ID#: bio.062072

MS TITLE: Osmoregulation in the estuarine diamond-backed terrapin across a broad range of naturally occurring salinities

AUTHORS: Amanda Williard; Jasmine Pierre; Brett Wilson

I have now reached a decision on the above manuscript.

The reviewer reports are shown at the bottom of this email or can be accessed, together with a copy of this decision letter, by going to:

As you will see, the reviewers raised a number of substantial criticisms that prevent me from accepting the paper at this stage.

They suggest, however, that a revised version might prove acceptable, if you can address their concerns. If you think that you can deal satisfactorily with the criticisms on revision, I would be pleased to see a revised manuscript. We would then return it to the reviewers.

At this stage, we also ask you to ensure your manuscript complies with our formatting guidelines. Provided you are able to fully address the referees' comments, we are positive about publication of your paper (we accept over 95% of revision submissions) and therefore hope you won't mind any extra work involved in reformatting your manuscript at this point.

Please ensure that you clearly highlight all changes made in the revised manuscript. Please avoid using 'Tracked changes' in Word files as these are lost in PDF conversion.

I should be grateful if you would also provide a point-by-point response detailing how you have dealt with the points raised by the reviewers in the 'Response to Reviewers' box. Please attend to all of the reviewers' comments. If you do not agree with any of their criticisms or suggestions please explain clearly why this is so.

Reviewer 1

Comments for the author

The new submission of this study is a much improved manuscript. They have acted appropriately on my main concern regarding high glucose levels indicating severe stress in some animals and removed those animals with very high glucose levels from the analyses. They have also shortened the Introduction and Discussion as I suggested. I therefore have no more comments except that I do agree with the other referee that it would be very useful for physiologists to see the data presented on linear scales rather than log transformed (which possibly could be moved to supplementary material).

Reviewer 2

Comments for the author

This is an interesting study on the osmoregulation of terrapins that attempts to understand the osmoregulation strategies employed in free-ranging animals and if these translate from previous laboratory data of captive animals. While I believe the data presented will be interesting and useful, there are two issues in the current manuscript presentation that need to be resolved before I can give an adequate assessment:

* First, there are two versions of manuscript in the file I was provided - the second one appearing in the document seems to be an earlier version with tracked changes. I have not read anything in the second version (with tracked changes) and based my review entirely on the clean version that appears first in the document. The exception to this is the figures + captions, which are only present in the second version with tracked changes, however within this section there also appears to be two versions of the figures so it's unclear which one I should be reviewing. This has made it somewhat difficult to fully interpret the manuscript.

* Second, I think the authors can make it easier for the readers to follow this work. Occasionally, claims are made without references, data is discussed without a reference to a figure or explanation, and the flow of content can feel slightly disjointed. This is not a major criticism of the manuscript because I think overall it is well written, but it is a reasonably complex topic, and I think a few small additions and clarifications would make it clearer for the read. Overall, I suggest the authors consider adding more 'signpost' sentences at the start of paragraphs. At times it is hard to follow the train of logic and whether the next paragraph is discussing the same point or a new point.

With these two points in mind, my general comments are:

* The manuscript has an unstated premise in that single time-point environmental conditions directly relate to blood osmolyte levels. This may be reasonable in this context, but it needs explaining/justification (see specific comment below for line 100).

* The manuscript posits to resolve uncertainty in how lab vs free-ranging terrapins osmoregulate, specifically that free-ranging terrapins have more energetically favourable options via behavioural adaptations (and not solely by changing osmolyte concentrations), however energy use was not measured and there are no citations provided that demonstrate the behavioural adaptations are less energetically costly. Indeed, if behaviour change is more energetically efficient then why do the individuals in the current study appear to be using non-behavioural mechanisms? I think the text on this point needs to be clearer because the manuscript presents it as a major driver of the study.

* I think much of the discussion is background information that can be placed in the introduction to better explain the importance and relevance of the work. This will also free up space in the discussion for 2-3 additional paragraphs to interpret the study's results in the context of previous work and conservation/management application, which I think currently are lacking.

Specific comments:

* Ln 56 - do they achieve this by drinking or through osmosis?

* Ln 88-91 - This needs a citation - how do we know it is more energy efficient?

* Ln 100 - I'm left wondering if there's a missing piece of the experimental logic here. The study assumes a link between environmental conditions (e.g. salinity) and blood levels of these variables, but how quickly do both change and how far can a terrapin move before it's blood levels will respond? For example, if the terrapin was collected during heavy rainfall, I assume the environmental salinity would change fairly rapidly (over the course of hours) but how quickly would the terrapin respond physiologically to this change? I'm sure the authors have considered this but don't seem to include it in the introduction.

* Ln 153- where does this conversion equation come from?

* Ln 161 - usually package names in R are italicised.

* Ln 170 - does the start of this paragraph relate to the previous paragraph's analysis? Or is this a new analysis?

* Ln 195 - which figure shows these data?

* Ln 200 - blood sodium levels? When describing lots of complex variables, I think the reader would benefit from very clear distinctions, including blood vs environmental variables.

* Discussion - can sub-headings be used? Throughout the discussion I find it difficult to know when one point has 'ended' and another discussion point has started, especially because sometimes interpretation of the present study's results is minimal.

* Ln 209-212 - but was there any interaction between body size and environmental variables for the response variables? E.g. urea increased with salinity and decreased with SCL, but how do we know that bigger terrapins don't just prefer less saline environments?

* Ln 213-241 - I'm not sure what the take home message in this paragraph is, and if Ln 234-236 is referring the present study. Please make it clear how the results of the present study support or refute these earlier studies. At the moment it feels a little bit like just a list of relevant but unstructured facts.

* Ln 243 - 245 - missing citation(s).

* Ln 245-246 - again missing citation.

* Ln 311-313 - again this claim is made without a reference: how do we know that behavioural adjustments are more energy efficient than osmoregulation strategies? The energetic cost of activity can be a significant portion of the overall energy budget.

* Ln 321-344 - I'm not sure what this paragraph adds to the discussion, it seems to be background information that should be in the introduction to explain why certain glucose levels were removed from analysis.

* Ln 345 - 371 - Again this does not seem like a conclusion to me but rather background information and justification for the study that belongs in the introduction.

* Figure 3 caption - variable names don't usually need to be capitalised.

* Other figures - I'm not sure which version of the figures I should be looking at, so will need to see this again on a cleaner version.

Reviewer's Responses to Questions

Experimental quality

Does each figure have the proper controls?

If 'No', please indicate reasons in Comments for Author box below.

Reviewer #1:

- Yes

Reviewer #2:

- No

Were the data analyzed using appropriate statistical tests?

If 'No', please indicate reasons in Comments for Author box below.

Reviewer #1:

- Yes

Reviewer #2:

- Yes

Reproducibility

Were experiments performed using adequate number of biological replicates?

If 'No', please indicate reasons in Comments for Author box below.

Reviewer #1:

- Yes

Reviewer #2:

- Yes

Does the methods section provide sufficient detail to permit reproducibility?

If 'No', please indicate reasons in Comments for Author box below.

Reviewer #1:

- Yes

Reviewer #2:

- Yes

Completeness

Are the manuscript's conclusions supported by the data?

If 'No', please indicate reasons in Comments for Author box below.

Reviewer #1:

- Yes

Reviewer #2:

- No

Scholarship

Do the authors cite and discuss the merits of data that would argue for and against their conclusion?

If 'No', please indicate reasons in Comments for Author box below.

Reviewer #1:

- Yes

Reviewer #2:

- No

Does the manuscript title & abstract accurately reflect the contents of the manuscript, without hyperbole?

If 'No', please indicate reasons in Comments for Author box below.

Reviewer #1:

- Yes

Reviewer #2:

- Yes

First revision

Author response to reviewers' comments

Comments from the Reviewers:

Reviewer 1: The new submission of this study is a much improved manuscript. They have acted appropriately on my main concern regarding high glucose levels indicating severe stress in some animals and removed those animals with very high glucose levels from the analyses. They have also shortened the Introduction and Discussion as I suggested. I therefore have no more comments except that I do agree with the other referee that it would be very useful for physiologists to see

the data presented on linear scales rather than log transformed (which possibly could be moved to supplementary material).

We thank the Reviewer for this feedback and are happy to hear that their concerns with the original manuscript were fully addressed. With regards to scale (linear vs log-transformed) for Figure 3, this Figure is showing the output of the model used to assess the significance of explanatory variables. Graphs that use the untransformed data show the same trends, but do not show model outputs. We provide descriptive statistics for the un-transformed data in Table 1 and provide raw data for environmental (salinity and temperature), morphological (straight carapace length [SCL]), and blood biochemistry data for each individual terrapin used in the statistical analysis in Table S1 (Supplementary Data) so that readers can see actual values for each blood variable for a given salinity or SCL. If Editors believe the raw data from Table S1 should be moved into the main manuscript in order for the raw data to be more accessible to the reader, we are happy to make this change. We can also include linear graphs if Reviewers and Editors feel that raw data are better presented in graph instead of Table form.

Reviewer 2: General comments:

This is an interesting study on the osmoregulation of terrapins that attempts to understand the osmoregulation strategies employed in free-ranging animals and if these translate from previous laboratory data of captive animals. While I believe the data presented will be interesting and useful, there are two issues in the current manuscript presentation that need to be resolved before I can give an adequate assessment:

* First, there are two versions of manuscript in the file I was provided - the second one appearing in the document seems to be an earlier version with tracked changes. I have not read anything in the second version (with tracked changes) and based my review entirely on the clean version that appears first in the document. The exception to this is the figures + captions, which are only present in the second version with tracked changes, however within this section there also appears to be two versions of the figures so it's unclear which one I should be reviewing. This has made it somewhat difficult to fully interpret the manuscript.

Figures for the Resubmission were provided in separate files, as requested in Instructions for Authors. The PDF that was compiled from the website and that I downloaded upon resubmission (bio.062072) had the revised version of the Figures on the last several pages (after the marked version of the original manuscript).

* Second, I think the authors can make it easier for the readers to follow this work. Occasionally, claims are made without references, data is discussed without a reference to a figure or explanation, and the flow of content can feel slightly disjointed. This is not a major criticism of the manuscript because I think overall it is well written, but it is a reasonably complex topic, and I think a few small additions and clarifications would make it clearer for the read. Overall, I suggest the authors consider adding more 'signpost' sentences at the start of paragraphs. At times it is hard to follow the train of logic and whether the next paragraph is discussing the same point or a new point.

Thank you for this suggestion. We have added a few sentences and incorporated sub-headings to try and improve the flow and clarity of the manuscript.

With these two points in mind, my general comments are:

* The manuscript has an unstated premise in that single time-point environmental conditions directly relate to blood osmolyte levels. This may be reasonable in this context, but it needs explaining/justification (see specific comment below for line 100).

Given the variability due to weather and tidal influences, it is difficult to document the exact environmental conditions experienced by each of the terrapins prior to sampling. Laboratory work indicates that it takes several days for blood variables to fully stabilize with exposure to a new, stable salinity (Bentley et al. 1967, Gilles-Baillien 1970), but under natural conditions terrapins are likely to experience frequent fluctuations in salinity and temperature. The blood biochemistry profiles we present are a snapshot of the physiological status of the terrapins in a dynamic environment to which terrapins are continually adjusting. That said, based on results from laboratory studies I think it is reasonable to assume that blood chemistry would reflect exposure to consistently low vs consistently high salinity, even if there was some hour-to-hour or day-to-day variability around the mean salinity. Despite the limitations of our point measurements, our study provides the first description of blood biochemistry for free-ranging terrapins over a broad range of environmental salinities and insight into the role of organic osmolytes and body size in terrapin osmoregulation.

* The manuscript posits to resolve uncertainty in how lab vs free-ranging terrapins osmoregulate, specifically that free-ranging terrapins have more energetically favourable options via behavioural adaptations (and not solely by changing osmolyte concentrations), however energy use was not measured and there are no citations provided that demonstrate the behavioural adaptations are less energetically costly. Indeed, if behaviour change is more energetically efficient then why do the individuals in the current study appear to be using non-behavioural mechanisms? I think the text on this point needs to be clearer because the manuscript presents it as a major driver of the study.

We have added references (Davenport and Macedo 1990, Davenport and Magill 1996, Harden et al. 2015, Williard et al. 2019) to clarify that the purported behavioral options used by terrapins (e.g. mud burial or basking, selective drinking behaviors, hypophagy) are energetically efficient. Previously published research on metabolic rate (VO_2) and water flux rates for terrapins during acute exposure to high salinity illustrates that evasive strategies to reduce salt influx (e.g. hypophagy and reduced drinking) contribute to osmotic homeostasis without the need for a significant increase in overall metabolic rate (Williard et al. 2019). We should note that radio-tracking studies of terrapin movements in coastal North Carolina indicate that they have small home ranges ($< 1 \text{ km}^2$) and long-distance movements between regions of low vs. high salinity have not been documented (Spivey 1998, Harden et al. 2012). Limited data ($N = 2$ female terrapins) from satellite telemetry studies along the Gulf coast of Florida show that movements of up to 50 km may occur (Lamont et al. 2021), but the authors related these movements to breeding and foraging behaviors, not specifically to osmoregulatory behaviors. The degree to which terrapins rely on movements to locate habitats with optimal/favorable salinity is certainly a topic worthy of additional study, but the published literature to date indicates that terrapins are reliant on modulation of food and water intake and shifts in the use of terrestrial (mud burial) vs aquatic habitats in their limited home ranges as behavioral means of osmoregulation.

* I think much of the discussion is background information that can be placed in the introduction to better explain the importance and relevance of the work. This will also free up space in the discussion for 2-3 additional paragraphs to interpret the study's results in the context of previous work and conservation/management application, which I think currently are lacking.

Both Reviewers for the original version of this manuscript suggested that the Introduction and Discussion should be substantially shortened and we followed this advice for the revision (Reviewer 1 - "While well-written, the manuscript is rather long given the rather small data set. I believe this may deter readers from finishing reading the manuscript and I strongly suggest that the authors consider shortening the speculative section." Reviewer 2 - "The introduction gives a good overview of the present knowledge of the osmoregulatory challenges and solutions for these terrapins. However, it is 4 pages long and more like a review of the subject and should be shortened substantially with focus of aspects directly relevant to the present study. Like the Introduction, the Discussion is clearly too long and includes aspects that are not directly relevant to the data obtained in the present study.").

I am hesitant to add several paragraphs to the Discussion, given the guidance from the original Reviewers to streamline the manuscript. The first paragraph of the Introduction highlights the

relevance of understanding physiological ecology of terrapins in the context of climate change and sea level rise, and the last paragraph of the Discussion specifically places the findings of our study (osmotic responses to changes in salinity) in the context of implications for adjusting/adapting to projected alterations in salinity profiles.

Specific comments:

* Ln 56 - do they achieve this by drinking or through osmosis?

By drinking. Clarification made in the text.

* Ln 88-91 - This needs a citation - how do we know it is more energy efficient?

Added the following citations to support this statement - Harden et al. 2015, Williard et al. 2019

* Ln 100 - I'm left wondering if there's a missing piece of the experimental logic here. The study assumes a link between environmental conditions (e.g. salinity) and blood levels of these variables, but how quickly do both change and how far can a terrapin move before it's blood levels will respond? For example, if the terrapin was collected during heavy rainfall, I assume the environmental salinity would change fairly rapidly (over the course of hours) but how quickly would the terrapin respond physiologically to this change? I'm sure the authors have considered this but don't seem to include it in the introduction.

Previous research to document the time course for response of blood variables has taken place under controlled laboratory conditions in which terrapins were moved from one set salinity to another; for example, from freshwater to full-strength seawater (Bentley et al. 1967, Gilles-Baillien 1970). These experiments illustrated that full acclimation of blood osmotic profile to a new, stable salinity treatment (after which no additional change in blood variables occurs) takes several days. Given the variability due to weather and tidal influences, it is difficult to document the exact field conditions experienced by each of the terrapins prior to sampling. With regards to rainfall, no terrapins were collected during or immediately following heavy rainfall (not safe for coastal boat work). All study sites had tidal influence and good mixing of water within the tidal creeks in which terrapins were collected. I can also confirm, based on our multi-year studies at these study sites, that dramatic, rapid changes in salinity (> 5 psu) are uncommon and typically associated with extreme weather patterns (e.g. hurricanes) - our sampling did not occur concurrent with or immediately following extreme weather events. Based on results from laboratory studies, we think it is reasonable to assume that blood chemistry would reflect exposure to consistently low vs consistently high salinity at the study sites, even if there was some hour-to-hour or day-to-day variability around the mean salinity. Despite the limitations of our point measurements, our study provides the first description of blood biochemistry for free-ranging terrapins over a broad range of environmental salinities and insight into the role of organic osmolytes and body size in terrapin osmoregulation.

* Ln 153- where does this conversion equation come from?

Standard clinical chemistry SI conversion factor
(<https://www.msdsmanual.com/multimedia/table/clinical-chemistry-si-conversion-factors>)

* Ln 161 - usually package names in R are italicised.

Package name italicized as requested

* Ln 170 - does the start of this paragraph relate to the previous paragraph's analysis? Or is this a new analysis?

Yes, it relates to the previous paragraph. The NMDS and PERMANOVA were used to identify explanatory variables to include in the linear models for each blood osmotic variable (lines 364 - 366 and lines 377 - 379 in revised, clean manuscript).

* Ln 195 - which figure shows these data?

The results of the PERMANOVA are not illustrated in graphic form, they are simply stated in the text. Figure 2 illustrates the results of the NMDS analysis and includes symbols to designate blood osmotic variables (+) relative to the ordination scores for individual terrapin blood profiles (\circ), as well as continuous explanatory variables overlaid onto the ordination plot as vectors (blue lines). (Lines 368 - 370 and legend for Figure 2)

* Ln 200 - blood sodium levels? When describing lots of complex variables, I think the reader would benefit from very clear distinctions, including blood vs environmental variables.

Specified blood sodium in revised manuscript.

* Discussion - can sub-headings be used? Throughout the discussion I find it difficult to know when one point has 'ended' and another discussion point has started, especially because sometimes interpretation of the present study's results is minimal.

The natural points to divide into subheadings are: Blood Osmolality and Organic Osmolytes, Body Size and Organic Osmolytes, Regulation of Inorganic Ions, Osmoregulation and Anthropogenic Stressors. I have done so in the revised manuscript, although this results in several subsections with just one paragraph. If the Reviewer and Editor find this helps with interpretation of study results I am fine with subsections. With regards to interpretation of results, the Discussion section was significantly longer in the original submission but both previous Reviewers suggested that we substantially shorten the Discussion.

* Ln 209-212 - but was there any interaction between body size and environmental variables for the response variables? E.g. urea increased with salinity and decreased with SCL, but how do we know that bigger terrapins don't just prefer less saline environments?

We checked for correlation of explanatory variables in our initial data exploration. There was no significant correlation between SCL and salinity ($r = -0.09$) or SCL and Tw ($r = 0.17$).

* Ln 213-241 - I'm not sure what the take home message in this paragraph is, and if Ln 234-236 is referring the present study. Please make it clear how the results of the present study support or refute these earlier studies. At the moment it feels a little bit like just a list of relevant but unstructured facts.

The previous paragraph states that our field results support results of laboratory studies that conclude use of organic osmolytes (urea) plays a role in the osmotic strategy of terrapins (lines 136 - 139 in revised, clean manuscript). The paragraph on lines 145 - 173 puts our results in the broader context of urea as an osmoeffector for vertebrates in a range of desiccating environments, including brackish to marine water, and highlights potential mechanisms for the increase in urea in terrapins based on published research with other species of turtles. Lines 166 - 168 refer to the increase in urea in terrapins that has now been documented in both laboratory (Gilles-Baillien 1970) and field studies (current manuscript). I added a few clarifying phrases and sentences to this paragraph to try and make the intent more clear.

* Ln 243 - 245 - missing citation(s).

Added citation - Costanzo et al. 2006

* Ln 245-246 - again missing citation.

General statement of environmental challenge, not a research finding. Same citation as in the previous and following sentences.

* Ln 311-313 - again this claim is made without a reference: how do we know that behavioural adjustments are more energy efficient than osmoregulation strategies? The energetic cost of activity can be a significant portion of the overall energy budget.

We have added text and references (Davenport and Macedo 1990, Davenport and Magill 1996, Harden et al. 2015, Williard et al. 2019) to clarify that the behavioral options used by terrapins are energetically efficient. The behavioral adjustments purported to contribute to osmoregulation (e.g. use of terrestrial habitats for basking or mud burial, reduction in foraging) do not involve an increase in activity, they actually involve a decrease in activity. Radiotracking shows that terrapins in North Carolina have very limited home ranges and published literature to date indicates that movements between terrestrial and aquatic habitats within these small home ranges provides behavioral means of osmoregulation.

* Ln 321-344 - I'm not sure what this paragraph adds to the discussion, it seems to be background information that should be in the introduction to explain why certain glucose levels were removed from analysis.

Previous reviewers requested that an explanation of the potential role of stress on blood biochemistry be provided, as well as an explanation of how we dealt with this. Lines 357 - 359 (revised, clean manuscript) explain our criteria for excluding individual terrapins from analysis based on blood glucose values and provides a reference for justification. We believe that a discussion of the effects of stress on corticosterone and aldosterone is relevant not just to the capture methodology and potential implications for interpretation of our data, but also for the impacts of anthropogenic stressors on terrapin physiology, as stated on lines 276 - 283 (revised, clean manuscript).

* Ln 345 - 371 - Again this does not seem like a conclusion to me but rather background information and justification for the study that belongs in the introduction.

Introduction to the study species conservation status and justification for study in the context of climate change and sea level rise is provided on lines 24 - 40 of the Introduction. The final paragraph of the Discussion places our findings in the context of documented changes in the terrapin's coastal habitats and implications for bioenergetics and the terrapin's osmotic strategy.

* Figure 3 caption - variable names don't usually need to be capitalised.

We made change as requested

* Other figures - I'm not sure which version of the figures I should be looking at, so will need to see this again on a cleaner version.

Figures for the Resubmission were provided in separate files, as requested in Instructions for Authors. The PDF that was compiled and that I downloaded upon resubmission (bio.062072) had the revised version of the Figures on the last several pages (after the marked version of the original manuscript). I assume this will also be the case for the revised version of this manuscript.

Second decision letter

MS ID#: bio.062072R1

MS TITLE: Osmoregulation in the estuarine diamond-backed terrapin across a broad range of naturally occurring salinities

AUTHORS: Amanda Williard; Jasmine Pierre; Brett Wilson

I've had a chance to read through your responses and resultant text edits, this morning. I am happy to tell you that your manuscript has been accepted for publication in Biology Open, pending our standard publication integrity checks. It was accepted on 02 Jun 2025. I'm fine with the current subheadings within the manuscript.